environmental science/biochemistry/environmental chemistry

coral reefs, calcareous sediments, skeletal development, terrestrial runoff, phosphate contamination

**Author for correspondence:**
Ko Yasumoto
e-mail: yasumoto@kitasato-u.ac.jp

# Phosphate bound to calcareous sediments hampers skeletal development of juvenile coral

Mariko Iijima[1], Jun Yasumoto[2], Akira Iguchi[1], Kiyomi Koiso[3], Sayaka Ushigome[3], Natsuki Nakajima[3], Yuko Kunieda[3], Takashi Nakamura[4], Kazuhiko Sakai[5], Mina Yasumoto-Hirose[6], Kanami Mori-Yasumoto[7], Nanami Mizusawa[3], Haruna Amano[3], Atsushi Suzuki[1], Mitsuru Jimbo[3], Shugo Watabe[3] and Ko Yasumoto[3]

[1]Geological Survey of Japan, National Institute of Advanced Industrial Science and Technology (AIST), 1-1-1 Higashi, Tsukuba, Ibaraki 305-8567, Japan
[2]Faculty of Agriculture, University of the Ryukyus, 1 Senbaru, Nishihara, Nakagusuku, Okinawa 903-0213, Japan
[3]School of Marine Biosciences, Kitasato University, 1-15-1 Kitasato, Minami, Sagamihara, Kanagawa 252-0373, Japan
[4]Faculty of Science, University of the Ryukyus, 1 Senbaru, Nishihara, Nakagusuku, Okinawa 903-0213, Japan
[5]Sesoko Station, Tropical Biosphere Research Center, University of the Ryukyus, Motobu, Okinawa, Japan
[6]Tropical Technology Plus, 12-75 Suzaki, Uruma, Okinawa 904-2234, Japan
[7]Faculty of Pharmaceutical Sciences, Tokyo University of Science, 2641 Yamazaki, Noda, Chiba 278-8510, Japan

MI, 0000-0002-5491-9309; AI, 0000-0002-8894-1977; KM-Y, 0000-0002-9718-0823; AS, 0000-0002-0266-5765; KY, 0000-0001-8706-8638

To test the hypothesis that terrestrial runoff affects the functions of calcareous sediments in coral reefs and hampers the development of corals, we analysed calcareous sediments with different levels of bound phosphate, collected from reef areas of Okinawajima, Japan. We confirmed that phosphate bound to calcareous sediments was readily released into ambient seawater, resulting in much higher concentrations of phosphorous in seawater from heavily polluted areas (4.3–19.0 µM as compared with less than 0.096 µM in natural ambient seawater). Additionally, we examined the effect of phosphate released from calcareous sediments on the development of *Acropora digitifera* coral juveniles. We found

that high phosphate concentrations in seawater clearly inhibit the skeletal formation of coral juveniles. Our results demonstrate that calcareous sediments in reef areas play a crucial role in mediating the impact of terrestrial runoff on corals by storing and releasing phosphate in seawater.

# 1. Introduction

The negative effects of terrestrial runoff on corals have been widely recognized [1–8]. Nutrient loads on coastal zones around coral reefs are variable, and mostly attributed to human activities [1,4,9–11]. Understanding the detailed relationship between local nutrient loads and coral growth is crucial for the conservation of coral reef ecosystems. In particular, since corals are oligotrophic, it has been suggested that an overload of nutrients such as dissolved inorganic nitrogen and phosphorus (DIN and DIP, respectively) poses a severe threat to coral reefs [12]. However, the processes through which DIN and DIP overload can harm corals are poorly understood: suggestions include both direct (e.g. inhibition of coral calcification [13] and larval settlement [1]) and indirect (e.g. proliferation of macroalgae [14], more frequent outbreaks of crown-of-thorns starfish deleterious effects [15]).

The Ryukyu Archipelago, in the southernmost region of Japan (electronic supplementary material, figure S1), contains coral reefs—mainly fringing reefs near coastal areas [16]. Okinawajima, the largest island in the Ryukyu Archipelago, is characterized by high human population and intense industrial and agricultural activity. Fringing reefs off Okinawajima are devastated by human-induced disturbance, such as red soil-derived terrigenous runoff [16,17]. Omija *et al.* reported the condition of corals and the nutrient concentration at 25 points around Okinawajima and found that corals did not grow well in terms of coral coverage and survival rate with a total nitrogen (TN) level of $0.1 \, \mathrm{mg \, l^{-1}}$ or more or a total phosphorus (TP) level of $0.01 \, \mathrm{mg \, l^{-1}}$ or more, although the reasons for this remain unclear [17].

While the water quality in coral reefs of the Ryukyu Archipelago has been well studied (e.g. [12,18]), little is known about the effects of coastal calcareous sediments on coral growth. It has been reported that phosphate bound to the calcareous sediments is released into seawater, which may contribute to the production of microphytobenthic communities, whereas phosphate dissolved in seawater is thought to be a minor source for the production of microphytobenthic communities [19]. As corals are sessile organisms whose larvae settle on the sea floor, the phosphate content of calcareous sediments in the reef area may directly affect coral settlement, because phosphate has a powerful calcium ion-chelating ability and is known to inhibit crystallization of calcium carbonate during biomineralization [20]. In this study, we hypothesize that phosphate from terrestrial runoff is (temporarily) stored in the calcareous sediment reservoir, and then released into seawater, possibly hampering coral development. To test this hypothesis, we used the *Acropora digitifera* juveniles (e.g. [21,22]), which enables us to evaluate the effects of environmental factors on coral calcification qualitatively with sufficient reproducibility.

# 2. Material and methods

## 2.1. Measurement of the concentration of phosphate released from calcareous sediments in seawater

Calcareous sediment samples (each about 10 g) were collected by gently scraping the seafloor surface with hands and wide-mouthed bottles from three sites off Okinawajima (electronic supplementary material, figure S1): one site in the north (Pt.1, Sesoko: 127°51′52.98″ E/26°38′4.02″ N), where corals are widespread inside a reef, and two sites off the southern coast, close to urban and agricultural areas (Pt.2, Odo: 127°42′30.90″ E/26°5′19.92″ N; Pt.3: Minatogawa: 127°45′33.00″ E/26°7′16.60″ N). Then, the sediment samples were dried, sieved to particle size fractions of less than 0.5 mm and 0.5–1 mm, and characterized with Fourier-transform infrared spectroscopy (FTIR) using a Nicolet iS5 spectrometer (Thermo Fisher Scientific K. K., Waltham, MA, USA). To determine TP bound to calcareous sediments, one gram of sieved sediments from each sample was dissolved in 5 ml of 6 M HCl for 24 h [19]. After centrifugation, the concentration of phosphate released in the solution was determined with molybdenum blue colourimetry [23], using a UV-1900 UV–visible spectrophotometer (Shimadzu, Kyoto, Japan).

For the coral rearing experiments (see below), samples of sieved sediment from the 0.5–1 mm particle size fraction were spread on Petri dishes and topped with 25 ml of filtered oligotrophic natural seawater collected from the ocean surface off the coast of Ogasawara Islands (filter pore size, 0.2 µm; nitrate, less than 0.35 µM; phosphate, less than 0.096 µM [13]). This oligotrophic seawater was replaced in the Petri dishes every 2 days. After 8 days of immersion (four changes of seawater) sediment samples were also replaced. In total, sediment samples were replaced four times during the rearing experiments. The concentration of phosphate in seawater used in the rearing experiments was measured though molybdenum blue colourimetry [23], using a UV-1900 UV–visible spectrophotometer (Shimadzu, Kyoto, Japan).

## 2.2. Rearing experiments of coral juveniles using calcareous sediments

Gametes of the scleractinian coral, *A. digitifera*, the dominant *Acropora* species in the Ryukyu Archipelago [24], were collected during a mass-spawning event in June 2019. Metamorphosis larvae were prepared from these gametes with 10 µM Hym-248 (Eurofins Genomics KK, Tokyo, Japan) at 25°C [25]. Juvenile corals were settled in 8-well chambered coverglasses (chamber size, $10 \times 10$ mm; type 5232-008-CS; AGC Techno Glass, Chiba, Japan) containing 100 µl of the oligotrophic natural seawater for each chamber. The chambered coverglasses containing juvenile corals were then placed onto Petri dishes (90 mm in diameter), where sieved calcareous sediments were filled with the oligotrophic natural seawater. Control samples of juvenile coral were reared without calcareous sediments. The formation of the basal plate of coral juveniles was observed 40 days after settlement using an AXIO Vert.A1 polarizing microscope (Carl Zeiss, Oberkochen, Germany). The calcification area ratio in juvenile corals was calculated by measuring the basal plate (aboral) area using *ImageJ* software (NIH, Bethesda, MD, USA). The oral side of the coral skeleton was observed with an M165FC stereomicroscope (Leica, Wetzlar, Germany) after light rinsing with ultrapure water. To evaluate changes in the skeleton surface in more detail, the juvenile corals reared for 40 days after settlement in calcareous sediments were rinsed with ultrapure water, coated with a platinum–palladium mixture (E-1045 sputter coater; Hitachi, Tokyo, Japan) and observed with a scanning electron microscope (SEM) model VE-9800 (Keyence, Osaka, Japan) operated at 1 kV.

The data were analysed using GraphPad Prism v.6.0 for Windows (San Diego, CA, USA), and statistically significant differences ($p < 0.05$) were calculated using Tukey's test.

# 3. Results and discussion

FTIR analysis revealed that the collected sediments were made of calcite and aragonite (electronic supplementary material, table S1). Measurement of phosphate released into solution after HCl treatment [19] showed that the bound phosphate load of these calcareous sediments was on average 4.9–14.2 µmol g$^{-1}$ dry wet (electronic supplementary material, table S2). These phosphates were easily released into ambient seawater, as also reported in a previous study [19]. Phosphate concentration in seawater brought to contact with the sampled sediment was 4.3–19.0 µM (figure 1)—much higher than that in the oligotrophic natural seawater ($p < 0.05$). The highest value of phosphate concentration, 19.0 µM, was detected in sediments from Minatogawa (Pt. 3: figure 1), possibly reflecting high pollution levels caused by terrestrial runoff from urban and agricultural areas. Inhibition of coral skeleton development was more severe at higher concentrations of phosphate released from sediment in seawater (figure 2). The calcification area ratio was lowest with sediments from Pt.2 (Odo) and Pt.3 (Mintogawa) followed by those from Pt.1 (Sesoko) ($p < 0.05$). Forty days post-settlement, all control samples of juvenile corals (reared without calcareous sediment) had formed septic and corallite wall structures with smooth surfaces. By contrast, the development of corallite and septic wall structures was inhibited in corals cultivated with calcareous sediment (figure 2*a*). SEM demonstrated that these damaged juveniles had unusual skeleton structures on their surface with elongated shapes and several holes (figure 3).

We have recently shown that phosphate over 5 µM clearly inhibited both *in vitro* aragonite CaCO$_3$ formation and *in vivo* aragonite formation in the skeleton of juvenile *A. digitifera* corals [13]. Thus, we supposed that phosphate bound to calcareous sediments from polluted areas would be readily released into ambient seawater, resulting in the inhibition of coral skeleton formation. Actually, the damaged juvenile coral skeleton cultivated with calcareous sediments of Pt.2 (Odo) and Pt.3 (Minatogawa) in the present study had appearances similar to those of the damaged skeleton

**Figure 1.** Phosphate concentration in seawater in contact with calcareous sediment from the sampling sites (Pt.1 Sesoko, Pt.2 Odo, Pt.3 Minatogawa), and in sediment-free ocean water (control). Bars indicate the standard deviation (1st, 2nd, 3rd: $n = 4$, 4th: $n = 3$). Phosphate concentration decreased after replacing seawater (water was changed every 2 days). After 8 days (four changes of seawater), sieved sediments were replaced with fresh sediment (in total, four changes of sediment per sample). The data were analysed using GraphPad Prism v.6.0 for Windows (San Diego, CA, USA), and statistically significant differences ($p < 0.05$) were calculated using Tukey's test. Results are indicated with means ± standard errors (s.e.). Letters represent statistical significance at the level of $p < 0.05$ in multiple comparisons using Tukey's test. The raw data is also available at Figshare (https://figshare.com/s/f13fa4107736f8bb5cf5).

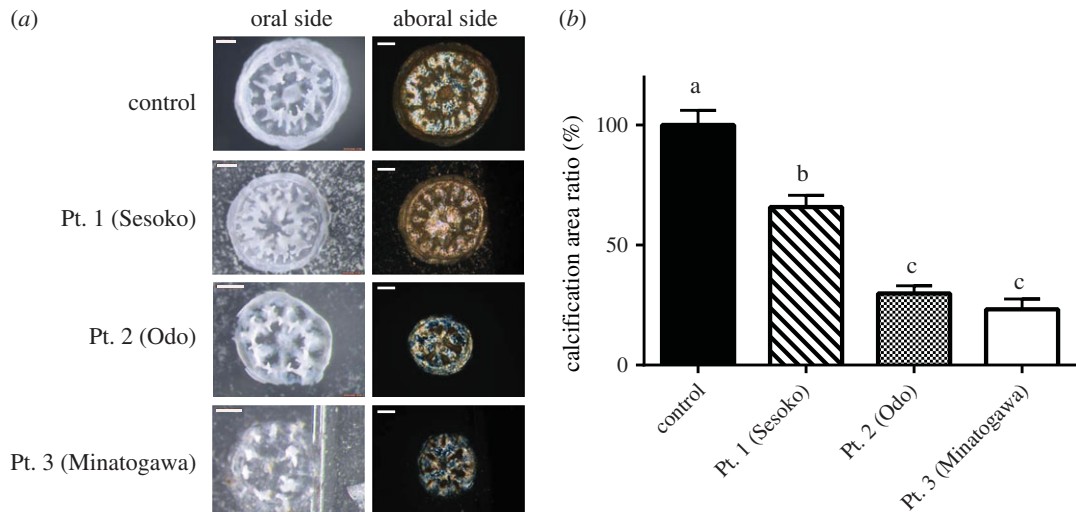

**Figure 2.** (*a*) Skeleton formation in *A. digitifera* juveniles 40 days post-settlement in the presence of calcareous sediments. Representative stereomicroscopic images of the oral side and representative polarizing microscopy images of the basal plate at the aboral side of juvenile corals reared with calcareous sediments Sesoko, Odo and Minatogawa (electronic supplementary material, figure S1). Control samples were reared without any sediment. Scale bars: 200 μm. (*b*) Calcification rates of *A. digitifera* juveniles (basal plates) in the presence of calcareous sediments. Different letters represent statistical significance at the level of $p < 0.05$ in multiple comparisons using Tukey's test. Results are indicated with means ± standard errors (s.e.) (control: $n = 23$; Pt. 1 (Sesoko): $n = 17$; Pt. 2 (Odo): $n = 11$; Pt. 3 (Minatogawa): $n = 9$). The raw data are also available at Figshare (https://figshare.com/s/bfa1358bfead1dacfa51).

cultivated with seawater containing over 5 μM phosphate mentioned above [13]. Although the other pollutants could also be released from the calcareous sediment, the characteristic inhibition of the juvenile coral skeletons led us to conclude that the skeleton inhibitions by the sediments were due to phosphate.

Tanaka *et al.* [22] also reported that high DIP values negatively affect the growth of juvenile corals, although they attributed this to the proliferation of benthic microalgae—an indirect effect. Calcareous

whole view          white square in whole view

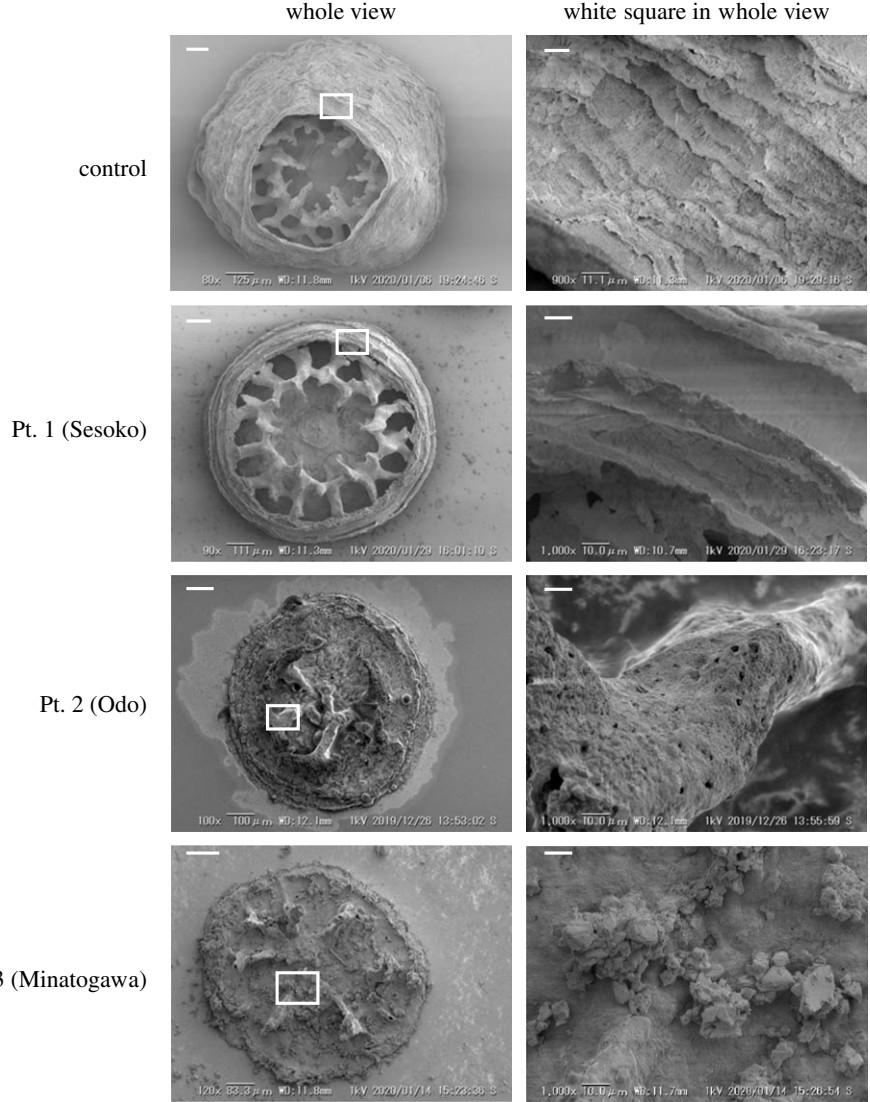

**Figure 3.** Representative SEM pictures showing skeleton surface of *A. digitifera* juveniles at 40 days post-settlement in the presence of calcareous sediments. Scale bars: 100 μm in whole views; 10 μm in the magnified pictures of the white square.

sediments, which are prone to adsorb phosphate, and function as both a sink and a source of orthophosphate in coastal environments [19]. We confirmed that calcareous sediments binding high TP showed high concentration of phosphate released into ambient seawater (figure 1 and electronic supplementary material, table S1). As sessile organisms, coral juveniles are likely to be strongly affected by phosphate released from calcareous sediments. Fast-growing *Acropora* spp. are rare in nutrient-rich areas, indicating that this species is particularly sensitive to eutrophication [1,26]. It has also been reported that higher phosphate concentrations result in unusual skeleton structures and decreased skeleton density of *Acropora* spp. due to inhibition of normal crystal formation [13,27]. Our results confirm these reports. Furthermore, our results show that clastic calcareous sediments are critical mediators of the phosphorous cycle, by storing terrigenous phosphate, and then releasing them into seawater, with deleterious effects on adjacent coral reefs. However, it is still unclear how sediment-bound phosphate is released under natural conditions. It is suggested that DIP concentrations are affected by adsorption–desorption equilibria at the ambient salinity [28]. Comprehending the temporal and spatial dynamics of sediment-bound phosphate under natural reef conditions will improve our understanding of the impact of terrestrial nutrients on corals and inform strategies for the protection of fragile coral reef ecosystems. These results also imply the potential threat of high phosphate-binding calcareous sediment on the initial recovery phase of the coral community following various disturbances [29].

# 4. Conclusion

The present study demonstrated that terrestrial phosphate is bound to calcareous sediments in coastal areas and the bound phosphate is readily released into ambient seawater, resulting in concentrations of phosphorous high in such seawater. These high phosphate concentrations were found to inhibit the skeletal formation of coral juveniles. Based on these data, we propose calcareous sediments in reef areas play a crucial role in mediating the impact of terrestrial runoff on corals by storing and releasing phosphate in seawater. We need more detailed survey research about phosphate bound to calcareous sediments near coasts for the managements of sustainable coral reef ecosystem services.

Ethics. We were allowed to harvest corals by Okinawa Prefectural permits (Numbers: 30-32, 31–43).

Data accessibility. All data required to repeat and validate the study are available in the electronic supplementary material.

Authors' contributions. M.I., K.Y., J.Y., A.I. conceived and designed the study; K.Y., K.K., S.U., N.N., Y.K., J.Y., A.I., T.N., K.S. collected samples and analysed the data; T.N., K.S., M.Y.-H., K.M.-Y., N.M., H.A., A.S., M.J., S.W. coordinated the study and helped draft the manuscript. All authors gave final approval for publication and agree to be held accountable for the work performed therein.

Competing interests. We have no competing interests.

Funding. This research was supported by the Environment Research and Technology Development Fund (grant no. JPMEERF20194007) of the Environmental Restoration and Conservation Agency of Japan, Grants-in-Aids from the Japan Society for the Promotion of Science (KAKENHI grant nos. 19K12310 and 20H03077), and Research Laboratory on Environmentally-conscious Developments and Technologies (E-code) at the National Institute of Advanced Industrial Science and Technology (AIST).

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
