## [Peer Review File · Royal Society Open Science]

Review History

RSOS-200407.R0 (Original submission)

Review form: Reviewer 1

Is the manuscript scientifically sound in its present form?

No

Are the interpretations and conclusions justified by the results?

No

Is the language acceptable?

Yes

Do you have any ethical concerns with this paper?

No

Have you any concerns about statistical analyses in this paper?

No

Recommendation?

Major revision is needed (please make suggestions in comments)

Comments to the Author(s)

Introduction:

This part is not well written and need to be rewritten according to the following comment: The authors have to write about the importance and ecosystem services of coral reefs, the impact of human activities on corals and the concentrate on sewage pollution and detergents. The goals should be clearer and new references is compulsory. I suggest the authors to cite the following related papers:

<https://www.sciencedirect.com/science/article/pii/S0045653517311633>

<https://www.sciencedirect.com/science/article/pii/S0048969718302262>

<https://www.sciencedirect.com/science/article/pii/S0048969718320916>

<https://www.sciencedirect.com/science/article/pii/S0269749119323887>

<https://www.sciencedirect.com/science/article/pii/S0045653520305907>

<https://www.sciencedirect.com/science/article/abs/pii/S0025326X18300869>

<https://www.sciencedirect.com/science/article/abs/pii/S0025326X18301905>

<https://www.sciencedirect.com/science/article/abs/pii/S0025326X13002920>

Materials and Methods:

This part is written poorly, the authors have to complete this section based on all preparation, instrumental analysis, quality assurance and control.

Statistical analysis has to be added.

Discussion is weak and needs to be improve.

There is no conclusion part and needs to be added.

Review form: Reviewer 2**Is the manuscript scientifically sound in its present form?**

No

Are the interpretations and conclusions justified by the results?

No

Is the language acceptable?

No

Do you have any ethical concerns with this paper?

No

Have you any concerns about statistical analyses in this paper?

Yes

Recommendation?

Major revision is needed (please make suggestions in comments)

Comments to the Author(s)

Review of Manuscript RSOS-200407

Title: Phosphates bound to calcareous sediments hamper skeletal development of juvenile coral.

General comments:

Potentially an interesting study - but see comments

Define specialist terms

Be more concise

Some rather vague sentences

Results: data needs to be clearly presented with means \pm SD/SE, ranges, number of replicates etc.

You do not appear to have conducted any statistics.

Introduction:

Define acronyms the first time you use them e.g. DIN, DIP, TN....there are many.

Lines 55-58 - rewrite this sentence - poor English.

Line 62 - define what you mean by high human population density. Numbers?

Line 66 - "corals did not grow well" - be more specific. Do you mean extension? Calcification? Disease? Tissue health? Etc.

Line 82 - this is a general science journal. Explain briefly to your (non expert, science) reader what the "primary polyp system" is. Then reference.

Materials and methods:

You need to give the reader enough information to duplicate your study.

Present in the correct order. E.g. Lines 102-105 - say: where they were collected from, how you collected them? grab?, how did you avoid contamination? Then that they were sieved.

Line 105 - define your "oligotrophic" range of nutrients in your experimental water. What was your baseline?

Was the water and sediment used for coral rearing experiments (line 102) collected from the same location - please clarify.

Line 117 an image would be useful of the experimental set up.

Line 118 - how big were the chambers? As you added them to petri dishes, they must be small.

Please give details.

Line 122 - this is not calcification rate - this appears to be surface area. Use units. Do you mean mm² = surface area. Calcification rates are measured in weight/area/time units e.g. mg cm⁻² y⁻¹.

Image J does not have the capacity to measure calcification.

Results and discussion

Line 130 Why are you using HCL - do you have reason to believe this mimics the natural environment? Explain method selection in methods & discussion.

135 you keep referring to "oligotrophic natural sea water" but have given no baseline data ranges to compare with. Please detail and define what you consider to be oligotrophic nutrient levels.

Line 135 - please give some results information - eg values of the "highest detected" in the text.

Line 140 explain what you mean by inhibited. An image here of controls and "inhibited" corals would be useful.

Line 141 explain "more severe"

Line 152-3 how do you know this is not a sedimentation / other pollution impact? Please explain.

Line 155 define "fragile" - explain/discuss.

Line 155 these results do not confirm "fragile skeletons" - if they do you have not presented the data.

Decision letter (RSOS-200407.R0)

Dear Dr Iijima:

Manuscript ID RSOS-200407 entitled "Phosphates bound to calcareous sediments hamper skeletal development of juvenile coral" which you submitted to Royal Society Open Science, has been reviewed. The comments from reviewers are included at the bottom of this letter.

In view of the criticisms of the reviewers, the manuscript has been rejected in its current form. However, a new manuscript may be submitted which takes into consideration these comments.

You'll note that a large number of citations are recommended for inclusion by one of the reviewers. Unfortunately the reviewer has neglected to provide a justification for the inclusion of so many references. Given the lack of explanation for their inclusion, the journal's Editors will not expect you to include these in your revised submission -- if any of the citations will add value to your manuscript, you are, of course, welcome to include those, but you should not (and are not expected to) add them all simply because the reviewer has asked you to do so.

Please note that resubmitting your manuscript does not guarantee eventual acceptance, and that your resubmission will be subject to peer review before a decision is made.

Your resubmitted manuscript should be submitted by 15-Nov-2020. If you are unable to submit by this date please contact the Editorial Office.

on behalf of Professor Stephen Hesselbo (Associate Editor)
openscience@royalsociety.org

Associate Editor Comments to Author (Professor Stephen Hesselbo):

Associate Editor: 1

Comments to the Author:

This study is potentially important and publishable in Royal Society Open Science, but requires substantial expansion in terms of methodological detail (sampling and experiment), statistics, referencing, discussion, and conclusions, as outlined by the referees. Therefore a revised manuscript would need to be treated as a new submission.

Reviewers' Comments to Author:

Reviewer: 1

Comments to the Author(s)

Introduction:

This part is not well written and need to be rewritten according to the following comment:
The authors have to write about the importance and ecosystem services of coral reefs, the impact of human activities on corals and the concentrate on sewage pollution and detergents. The goals should be clearer and new references is compulsory. I suggest the authors to cite the following related papers:

<https://www.sciencedirect.com/science/article/pii/S0045653517311633>

<https://www.sciencedirect.com/science/article/pii/S0048969718302262>

<https://www.sciencedirect.com/science/article/pii/S0048969718320916>

<https://www.sciencedirect.com/science/article/pii/S0269749119323887>

<https://www.sciencedirect.com/science/article/pii/S0045653520305907>

<https://www.sciencedirect.com/science/article/abs/pii/S0025326X18300869>

<https://www.sciencedirect.com/science/article/abs/pii/S0025326X18301905>

<https://www.sciencedirect.com/science/article/abs/pii/S0025326X13002920>

Materials and Methods:

This part is written poorly, the authors have to complete this section based on all preparation, instrumental analysis, quality assurance and control.

Statistical analysis has to be added.

Discussion is weak and needs to be improve.

There is no conclusion part and needs to be added.

Reviewer: 2

Comments to the Author(s)

Review of Manuscript RSOS-200407

Title: Phosphates bound to calcareous sediments hamper skeletal development of juvenile coral.

General comments:

Potentially an interesting study – but see comments

Define specialist terms

Be more concise

Some rather vague sentences

Results: data needs to be clearly presented with means \pm SD/SE, ranges, number of replicates etc.

You do not appear to have conducted any statistics.

Introduction:

Define acronyms the first time you use them e.g. DIN, DIP, TN....there are many.

Lines 55-58 – rewrite this sentence – poor English.

Line 62 – define what you mean by high human population density. Numbers?

Line 66 – “corals did not grow well” – be more specific. Do you mean extension? Calcification? Disease? Tissue health? Etc.

Line 82 – this is a general science journal. Explain briefly to your (non expert, science) reader what the “primary polyp system” is. Then reference.

Materials and methods:

You need to give the reader enough information to duplicate your study.

Present in the correct order. E.g. Lines 102-105 – say: where they were collected from, how you collected them ?grab?, how did you avoid contamination? Then that they were sieved.

Line 105 – define your “oligotrophic” range of nutrients in your experimental water. What was your baseline?

Was the water and sediment used for coral rearing experiments (line 102) collected from the same location – please clarify.

Line 117 an image would be useful of the experimental set up.

Line 118 – how big were the chambers? As you added them to petri dishes, they must be small.

Please give details.

Line 122 - this is not calcification rate - this appears to be surface area. Use units. Do you mean mm² = surface area. Calcification rates are measured in weight/area/time units e.g. mg cm⁻² y⁻¹. Image J does not have the capacity to measure calcification.

Results and discussion

Line 130 Why are you using HCL - do you have reason to believe this mimics the natural environment? Explain method selection in methods & discussion.

135 you keep referring to "oligotrophic natural sea water" but have given no baseline data ranges to compare with. Please detail and define what you consider to be oligotrophic nutrient levels.

Line 135 - please give some results information - eg values of the "highest detected" in the text.

Line 140 explain what you mean by inhibited. An image here of controls and "inhibited" corals would be useful.

Line 141 explain "more severe"

Line 152-3 how do you know this is not a sedimentation / other pollution impact? Please explain.

Line 155 define "fragile" - explain/discuss.

Line 155 these results do not confirm "fragile skeletons" - if they do you have not presented the data.

Author's Response to Decision Letter for (RSOS-200407.R0)

See Appendix A.

RSOS-201214.R0

Review form: Reviewer 3

Is the manuscript scientifically sound in its present form?

No

Are the interpretations and conclusions justified by the results?

No

Is the language acceptable?

Yes

Do you have any ethical concerns with this paper?

No

Have you any concerns about statistical analyses in this paper?

No

Recommendation?

Major revision is needed (please make suggestions in comments)

Comments to the Author(s)

I have a problem with the methodology used. The sediments released phosphate by they also probably released many other pollutants that have not been measured here. That means that the bad calcification process could have been induced here by another compound than phosphate. I

think that this study should at least be completed by the same experiment using clean seawater only enriched with phosphate at different concentrations to verify that phosphate is mainly responsible of bad calcification.

I don't know to what extent the word "phosphates" has to be written with an "s"? Is it because of the different forms of the phosphate ions or the different origins of phosphate?
Same remark for "nitrates" in the Discussion.

L. 53-55 : "...it has been suggested that nutrients such as dissolved inorganic nitrogen and phosphorus (DIN and DIP, respectively) pose a severe threat to coral reefs [12]". DIN and DIP are necessary for coral to thrive, I would have written "DIN and DIP overload pose a severe...".

L. 60-61: "The Ryukyu Archipelago, in the northernmost region of Japan (Fig S1), contains coral reefs". Isn't the Ryukyu Archipelago in the south of Japan?

L.152-153: Could you please add Pt 1, 2 or 3 next to the sampling site to help reading?

L.161: Replace "attributed to this to" by "attributed this to".

Figure S1: Could you please make the scale bar more visible?

Decision letter (RSOS-201214.R0)

Dear Dr Iijima

The Editors assigned to your paper RSOS-201214 "Phosphates bound to calcareous sediments hamper skeletal development of juvenile coral" have now received comments from reviewers and would like you to revise the paper in accordance with the reviewer comments and any comments from the Editors. Please note this decision does not guarantee eventual acceptance.

Please submit your revised manuscript and required files (see below) no later than 21 days from today's (ie 21-Jan-2021) date. Note: the ScholarOne system will 'lock' if submission of the revision is attempted 21 or more days after the deadline. If you do not think you will be able to meet this deadline please contact the editorial office immediately.

on behalf of Professor Stephen Hesselbo (Associate Editor)

Associate Editor Comments to Author (Professor Stephen Hesselbo):

Comments to the Author:

The revised manuscript addresses well the comments made by previous reviewers.

The new Reviewer 1 makes the very good point that the reported experiments do not determine whether phosphate or some other pollutant is the primary factor negatively affecting coral growth. The reviewer suggests a simple experiment to determine whether this is the case or not. It may be that such experiments have already been carried out and reported, but this is not clear in the current manuscript. This point should be addressed before the manuscript is published.

Reviewer comments to Author:

Reviewer: 3

Comments to the Author(s)

I have a problem with the methodology used. The sediments released phosphate by they also probably released many other pollutants that have not been measured here. That means that the bad calcification process could have been induced here by another compound than phosphate. I think that this study should at least be completed by the same experiment using clean seawater only enriched with phosphate at different concentrations to verify that phosphate is mainly responsible of bad calcification.

I don't know to what extent the word "phosphates" has to be written with an "s"? Is it because of the different forms of the phosphate ions or the different origins of phosphate?

Same remark for "nitrates" in the Discussion.

L. 53-55 : "...it has been suggested that nutrients such as dissolved inorganic nitrogen and phosphorus (DIN and DIP, respectively) pose a severe threat to coral reefs [12]". DIN and DIP are necessary for coral to thrive, I would have written "DIN and DIP overload pose a severe...".

L. 60-61: "The Ryukyu Archipelago, in the northernmost region of Japan (Fig S1), contains coral reefs". Isn't the Ryukyu Archipelago in the south of Japan?

L.152-153: Could you please add Pt 1, 2 or 3 next to the sampling site to help reading?

L.161: Replace “attributed to this to” by “attributed this to”.

Figure S1: Could you please make the scale bar more visible?

===PREPARING YOUR MANUSCRIPT===

===PREPARING YOUR REVISION IN SCHOLARONE===

<https://royalsociety.org/journals/authors/author-guidelines/#supplementary-material> to include a suitable title and informative caption. An example of appropriate titling and captioning may be found at https://figshare.com/articles/Table_S2_from_Is_there_a_trade-off_between_peak_performance_and_performance_breadth_across_temperatures_for_aerobic_sc_ope_in_teleost_fishes_/3843624.

Author's Response to Decision Letter for (RSOS-201214.R0)

See Appendix B.

Decision letter (RSOS-201214.R1)

Dear Dr Iijima,

It is a pleasure to accept your manuscript entitled "Phosphate bound to calcareous sediments hamper skeletal development of juvenile coral" in its current form for publication in Royal Society Open Science.

Best regards,

on behalf of Professor Stephen Hesselbo (Associate Editor)
openscience@royalsociety.org

Associate Editor Comments to Author (Professor Stephen Hesselbo):

The authors have now addressed all reviewer comments.

Appendix A

Reviewers' Comments to Author:

Reviewer: 1

Comments to the Author(s)

Introduction:

This part is not well written and need to be rewritten according to the following comment: The authors have to write about the importance and ecosystem services of coral reefs, the impact of human activities on corals and the concentrate on sewage pollution and detergents. The goals should be clearer and new references is compulsory. I suggest the authors to cite the following related papers:

<https://www.sciencedirect.com/science/article/pii/S0045653517311633>

<https://www.sciencedirect.com/science/article/pii/S0048969718302262>

<https://www.sciencedirect.com/science/article/pii/S0048969718320916>

<https://www.sciencedirect.com/science/article/pii/S0269749119323887>

<https://www.sciencedirect.com/science/article/pii/S0045653520305907>

<https://www.sciencedirect.com/science/article/abs/pii/S0025326X18300869>

<https://www.sciencedirect.com/science/article/abs/pii/S0025326X18301905>

<https://www.sciencedirect.com/science/article/abs/pii/S0025326X13002920>

Response: We selected a few references mentioned above and added them in the revised manuscript.

Materials and Methods:

This part is written poorly, the authors have to complete this section based on all preparation, instrumental analysis, quality assurance and control.

Statistical analysis has to be added.

Response: We added details of our experiments as well as statistical analysis in the revised manuscript as suggested.

Discussion is weak and needs to be improve.

Response: We added the SEM observation and discussed in the revised manuscript more about unusual structures due to the presence of high phosphates in the culture medium.

There is no conclusion part and needs to be added.

Response: we added conclusion to the revised manuscript as suggested.

Reviewer: 2

Comments to the Author(s)

Review of Manuscript RSOS-200407

Title: Phosphates bound to calcareous sediments hamper skeletal development of juvenile coral.

General comments:

Potentially an interesting study – but see comments

Define specialist terms

Be more concise

Some rather vague sentences

Results: data needs to be clearly presented with means \pm SD/SE, ranges, number of replicates etc. You do not appear to have conducted any statistics.

Response: we conducted statistical analysis and the methods were added to the revised manuscript. Detailed data are described in the legends of Fig. 1 and 2.

Introduction:

1. Define acronyms the first time you use them e.g. DIN, DIP, TN....there are many.

Response: we explained abbreviations in the revised manuscript as follows.

“In particular, since corals are oligotrophic, it has been suggested that nutrients such as dissolved inorganic nitrogen and phosphorus (DIN and DIP, respectively) pose a severe threat to coral reefs [12].”

“TN: Total nitrogen, TP : Total phosphorous”

2. Lines 55-58 – rewrite this sentence - poor English.

Response: we revised the sentence by defining acronyms for the first use as mentioned above.

3. Line 62 – define what you mean by high human population density. Numbers?

Response: we revised the sentence as follows.

“particularly high human population density” to “high human population”

4. Line 66 – “corals did not grow well” – be more specific. Do you mean extension? Calcification? Disease? Tissue health? Etc.

Response: we added “in terms of coral coverage and survival rate” in the revised manuscript.

5. Line 82 – this is a general science journal. Explain briefly to your (non expert, science) reader what the “primary polyp system” is. Then reference.
Materials and methods:

Response: we changed from “primary polyp system” to “juveniles”.

6. You need to give the reader enough information to duplicate your study.

Response: we provided experimental details such as collection of calcareous sediment samples, rearing system using 8-well chambered coverglasses and statistical analysis as suggested.

7. Lines 102-105 – say: where they were collected from, how you collected them ?grab?, how did you avoid contamination? Then that they were sieved.

Response: we revised the sentence to “Calcareous sediment samples (each about 10 g) were collected by gently scraping the seafloor surface with hands and wide-mouthed bottles from three sites off Okinawajima”.

8. Line 105 – define your “oligotrophic” range of nutrients in your experimental water. What was your baseline?

Response: we referred to our previous report (Iijima et al., 2019) and rephrased to “oligotrophic natural seawater collected from the ocean surface off the coast of Ogasawara Islands (filter pore size, 0.2 μm ; nitrate, < 0.35 μM ; phosphate, < 0.096 μM [13])”.

9. Was the water and sediment used for coral rearing experiments (line 102) collected from the same location – please clarify.

Response: We used the oligotrophic seawater collected from Ogasawara Islands in the coral rearing experiments and clarified this in the sentence “This oligotrophic seawater was replaced in the Petri dishes every two days

10. Line 117 an image would be useful of the experimental set up.

Response: we added chamber size and type of the product in the revised manuscript as follows: Juvenile corals were settled in 8-well chambered coverglasses (chamber size, 10 mm x 10 mm; type 5232-008-CS; AGC Techno Glass, Chiba, Japan) containing 100 μ L the oligotrophic natural seawater for each chamber. The chambered coverglasses containing juvenile corals were then inserted to Petri dishes (90 mm in diameter), where sieved calcareous sediments were filled with the oligotrophic natural seawater.

11. Line 118 – how big were the chambers? As you added them to petri dishes, they must be small. Please give details.

Response: we gave details as described in the above section.

12. Line 122 – this is not calcification rate – this appears to be surface area. Use units. Do you mean mm^2 = surface area. Calcification rates are measured in weight/area/time units e.g. $\text{mg cm}^{-2} \text{y}^{-1}$. Image J does not have the capacity to measure calcification.

Results and discussion

Response: we changed as follows: “The calcification rate” to “The calcification area ratio”.

13. Line 130 Why are you using HCL – do you have reason to believe this mimics the natural environment? Explain method selection in methods & discussion.

Response: we followed the method of Suzumura et al. (19) and added this reference number in the revised manuscript. Actually they determined phosphorus adsorbed on the

calcareous sediments of a coral reef by this method.

14. you keep referring to “oligotrophic natural sea water” but have given no baseline data ranges to compare with. Please detail and define what you consider to be oligotrophic nutrient levels.

Response: as in our response to your inquiry of No. 8, we used oligotrophic natural sea water off Ogasawara Islands located in mid Pacific Ocean. We referred to our previous report about nitrate of $< 0.35 \mu\text{M}$ and phosphate of $0.096 \mu\text{M}$, the description of which was added to the revised manuscript.

15. Line 135 – please give some results information – eg values of the “highest detected” in the text.

Response: we rephrased the sentence to “The highest value of phosphate concentration, $19.0 \mu\text{M}$, was detected in sediments from Minatogawa (Pt. 3: Fig. 1), “ in the revised manuscript as suggested.

16. Line 140 explain what you mean by inhibited. An image here of controls and “inhibited” corals would be useful.

Response: we added the results obtained by SEM in the revised manuscript and rephrased from “Inhibition of coral skeleton development more severe at higher concentrations of phosphates released from sediment in seawater (Fig. 1, 2)” to “SEM demonstrated that these damaged juveniles had unusual skeleton structures on their surface with elongated shapes and several holes (Fig. 3).”

17. Line 141 explain “more severe”

Response: as in the previous response, we added the results obtained by SEM in the revised manuscript.

18. Line 152-3 how do you know this is not a sedimentation / other pollution impact? Please explain.

Response: we deleted “polluted” and added “nutrient-rich” in the revised manuscript.

19. Line 155 define “fragile” – explain/discuss.

Response: we rephrased to “unusual skeleton structures and decreased skeleton density of Acropora spp. due to inhibition of normal crystal formation”.

20. Line 155 these results do not confirm “fragile skeletons” – if they do you have not presented the data.

Response: we deleted “fragile skeletons” and added related data reported previously in the revised manuscript.

Journal Name: Royal Society Open Science

Journal Code: RSOS

Online ISSN: 2054-5703

Journal Admin Email: openscience@royalsociety.org

Journal Editor: Andrew Dunn

Journal Editor Email: openscience@royalsociety.org

MS Reference Number: RSOS-200407

Article Status: REJECTED

MS Dryad ID: RSOS-200407

MS Title: Phosphates bound to calcareous sediments hamper skeletal development of juvenile coral

MS Authors: Iijima, Mariko; Yasumoto, Jun; Iguchi, Akira; Koiso, Kiyomi; Ushigome, Sayaka; Nakajima, Natsuki; Kunieda, Yuko; Nakamura, Takashi; Sakai, Kazuhiko; Yasumoto-Hirose, Mina; Mori-Yasumoto, Kanami; Mizusawa, Nanami; Amano, Haruna; Suzuki, Atsushi; Jimbo, Mitsuru; Watabe, Shugo; Yasumoto, Ko

Contact Author: Mariko Iijima

Contact Author Email: m.ijijima@aist.go.jp

Contact Author Address 1:

Contact Author Address 2:

Contact Author Address 3:

Contact Author City: Tsukuba

Contact Author State:

Contact Author Country: Japan

Contact Author ZIP/Postal Code: 305-8567

Keywords: Coral reefs, Skeletal development, Phosphate contamination, Terrestrial runoff

Abstract: To test the hypothesis that terrestrial runoff affects the functions of calcareous sediments in coral reefs and hampers the development of corals, we analysed calcareous sediments with different levels of bound phosphates, collected from reef areas offshore of Okinawajima, Japan. We confirmed that phosphates bound to calcareous sediments were readily released into ambient seawater, resulting in much higher concentrations of phosphorous in seawater from heavily polluted areas (4.3 to 19.0 μM as compared with $< 0.096 \mu\text{M}$ in natural ambient seawater). Additionally, we examined the effect of phosphates released from calcareous sediments on the development of *Acropora digitifera* coral juveniles. We found that high phosphate concentrations in seawater clearly inhibit skeletal formation of coral juveniles. Our results demonstrate that calcareous sediments in reef areas play a crucial role in mediating the impact of terrestrial runoff on corals by storing and releasing phosphate in seawater.

EndDryadContent

Appendix B

Reviewer comments to Author:

Reviewer: 3

Comments to the Author(s)

I have a problem with the methodology used. The sediments released phosphate by they also probably released many other pollutants that have not been measured here. That means that the bad calcification process could have been induced here by another compound than phosphate. I think that this study should at least be completed by the same experiment using clean seawater only enriched with phosphate at different concentrations to verify that phosphate is mainly responsible of bad calcification.

Response: we revised the sentence in page 6 line 161- as follows.

“We have recently shown that phosphate over 5 μM clearly inhibited both *in vitro* aragonite CaCO_3 formation and *in vivo* aragonite formation in the skeleton of juvenile *A. digitifera* corals [13]. Thus, we supposed that phosphate bound to calcareous sediments from polluted areas would be readily released into ambient seawater, resulting in the inhibition of coral skeleton formation. Actually, the damaged juvenile coral skeleton cultivated with calcareous sediments of Pt.2 (Odo) and Pt.3 (Minatogawa) in the present study had appearances similar to those of the damaged skeleton cultivated with seawater containing over 5 μM phosphate mentioned above [13]. Although the other pollutants could be also released from the calcareous sediment, the characteristic inhibition of the juvenile coral skeletons led us to conclude that the skeleton inhibitions by the sediments were due to phosphate.”

I don't know to what extent the word “phosphates” has to be written with an “s”? Is it because of the different forms of the phosphate ions or the different origins of phosphate? Same remark for “nitrates” in the Discussion.

Response: we changed the word “phosphates” and “nitrates” to “phosphate” and “nitrate” in our manuscript.

L. 53-55 : “...it has been suggested that nutrients such as dissolved inorganic nitrogen and phosphorus (DIN and DIP, respectively) pose a severe threat to coral reefs [12]”. DIN and DIP are necessary for coral to thrive, I would have written “DIN and DIP overload pose a severe...”.

Response: we added the word “overload” in page 3 line 54 to 56 as follows

“In particular, since corals are oligotrophic, it has been suggested that an overload of nutrients such as

dissolved inorganic nitrogen and phosphorus (DIN and DIP, respectively) poses a severe threat to coral reefs [12]. However, the processes through which DIN and DIP overload can harm corals are poorly understood: suggestions include both direct (e.g. inhibition of coral calcification [13] and larval settlement [1]) and indirect (e.g. proliferation of macroalgae [14], more frequent outbreaks of crown-of-thorns starfish deleterious effects [15].”

L. 60-61: “The Ryukyu Archipelago, in the northernmost region of Japan (Fig S1), contains coral reefs”. Isn’t the Ryukyu Archipelago in the south of Japan?

Response: we changed the sentence “northernmost region of Japan” to “southernmost region of Japan” in page 3 line 61.

L.152-153: Could you please add Pt 1, 2 or 3 next to the sampling site to help reading?

Response: we added Pt 1, 2 and 3 in page 6 line 154-155.

L.161: Replace “attributed to this to” by “attributed this to”.

Response: we revised the sentence in page 6 line 175

Figure S1: Could you please make the scale bar more visible?

Response: we revised the scale bar in Fig.S1.